# The Impact of Aquatic Exercise Programs on the Intima-Media thickness of the Carotid Arteries, Hemodynamic Parameters, Lipid Profile and Chemokines of Community-Dwelling Older Persons: A Randomized Controlled Trial

**DOI:** 10.3390/ijerph19063377

**Published:** 2022-03-13

**Authors:** Carlos Farinha, Hélder Santos, João Serrano, Bárbara Oliveiros, Fernanda M. Silva, Márcio Cascante-Rusenhack, Ana Maria Teixeira, José Pedro Ferreira

**Affiliations:** 1Research Unit for Sport and Physical Activity, CIDAF, UID/PTD/04213/2019, University of Coimbra, 3040-248 Coimbra, Portugal; geral.fernandasilva@gmail.com (F.M.S.); marciocascante@gmail.com (M.C.-R.); ateixeira@fcdef.uc.pt (A.M.T.); jpferreira@fcdef.uc.pt (J.P.F.); 2Municipality of Sertã, 6100-738 Sertã, Portugal; 3Coimbra School of Health Technology-IPC, 3046-854 Coimbra, Portugal; heldersantos98@gmail.com; 4Sport, Health & Exercise Research Unit (SHERU), Polytechnic Institute of Castelo Branco, 6000-266 Castelo Branco, Portugal; j.serrano@ipcb.pt; 5Faculty of Medicine, University of Coimbra, 3000-370 Coimbra, Portugal; boliveiros@fmed.uc.pt; 6School of Physical Education and Sports, University of Costa Rica (UCR), San José 11501-2060, Costa Rica

**Keywords:** physical exercise, aquatic environment, hydrogymnastics, ageing, intima-media thickness of the carotid arteries, hemodynamic parameters, blood pressure, lipid profile, MDP-1α, MCP-1

## Abstract

Scientific evidence has shown that physical exercise is an effective way of improving several cardiovascular disease markers. However, few studies have tested its effectiveness when performed in aquatic environments. The purpose of this study was to test the impact of different aquatic exercise programs on the intima-media thickness of carotid arteries (IMT) and hemodynamic and biochemical markers of cardiovascular diseases in community-dwelling older persons. A total of 102 participants were randomly allocated into four groups: an aerobic exercise group (AerG) (*n* = 25, 71.44 ± 4.84 years); an aerobic interval group (IntG) (*n* = 28, 72.64 ± 5.22 years); a combined group (ComG) (*n* = 29, 71.90 ± 5.67 years); and a control group (CG) (*n* = 20, 73.60 ± 5.25 years). The AerG, IntG, and ComG participants took part in three different aquatic exercise programs for 28 weeks. The CG participants maintained their usual routines. All participants were evaluated for IMT, blood pressure, lipid profile, and MCP-1 and MIP-1α chemokines, pre- and post-intervention. Significant differences were found in the AerG for diastolic diameter (DD), in the IntG for peak systolic velocity (PSV), and in the ComG for DD and end-diastolic velocity (EDV). Regarding blood pressure, significant differences were found in AerG for systolic blood pressure (SBP) and diastolic blood pressure (DBP); in IntG for DBP; and in ComG for SBP, DBP, and heart rate (HR). Significant differences were found in the AerG and IntG for glucose (GLU). Lower plasma levels of monocyte chemoattractant protein-1 (MCP-1) and macrophage inflammatory protein (MIP-1α) were found in the AerG and in the ComG for MCP-1 after the intervention. Aquatic physical exercise appears to improve cardiovascular health, regardless of the type of the program adopted. Aerobic programs (combined and continuous aerobic exercises) seemed to have a more beneficial effect in reducing important cardiovascular risk markers.

## 1. Introduction

Cardiovascular diseases are a group of heart and blood vessel diseases that are responsible for about 31% of the world’s deaths, 85% of which are caused by heart attacks and strokes [1]. According to the same source, it is estimated that 80% of these deaths could be avoided by controlling the main risk factors: smoking, unbalanced diets and physical inactivity. Evidence shows that about 150 min of moderate-intensity physical activity per week can contribute to a reduction in the risk of ischemic heart disease, risk of stroke, and hypertension by approximately 30% [1]. The ageing process is associated with cardiovascular changes that affect arterial function, leading to an increased risk of developing cardiovascular disease [2].

Cardiovascular diseases can be predicted through hemodynamic cardiovascular parameters and cardiovascular risk biomarkers. In relation to hemodynamic parameters, the intima-media thickness of the carotid arteries (IMT) is considered an early marker of coronary artery disease and also a risk factor for other types of cardiovascular disease and stroke [3]. The IMT is defined as the length between the inner layer of the carotid artery tunica intima and the inner layer of the common carotid artery tunica media. Increased IMT was positively and strongly associated with age [4], with values between 0.40 mm and 0.89 mm for IMT considered normal, thickening when values vary between 0.90 mm and 1.40 mm, and when wider than 1.40 mm, there is possibility that the carotid arteries contain plaques [5,6]. Other important markers are systolic and diastolic diameter and peak systolic and diastolic velocity. Arteries lose elasticity with ageing, thus increasing systolic blood pressure (which will consequently cause an increase in the systolic diameter) and blood flow velocity, usually leading to turbulent blood flow and loosening atheromatous plaques, which can consequently lead to obstruction of the vessels [7]. According to the same author, the systolic diameter (SD) corresponds to the highest point of each pulse, when the vessels are subjected to the highest pressure, and the diastolic diameter (DD) corresponds to the lowest arterial pressure point to which the vessels are subjected.

Concerning cardiovascular parameters, arterial hypertension is considered an early marker of the development of atherosclerotic disease, being considered a high-prevalence cardiovascular risk factor associated with endothelial dysfunction [8]. Atherosclerosis starts due to damage to the endothelial lining of large vessels triggered by pathogenic factors such as hyperlipemia and hemodynamic shear stress. It is a pathology that develops and progresses over a long period of time and consequently can lead to serious damage of the vascular tissue [9]. The development of atherosclerosis can consequently lead to ischemic heart disease, stenosis of the carotid arteries, and chronic lower limb ischemia, among others [10].

Regarding biochemical markers, ageing also leads to a deterioration of the lipid and glucose profile, increasing the levels of total cholesterol (TC), low-density lipoprotein (LDL), triglycerides (TG), and fasting blood glucose (GLU). When those elements are uncontrolled, they are directly related to cardiovascular complications [11]. Chemokines also play a fundamental role in the development of cardiovascular diseases and play a key role in the pathogenesis of inflammatory and autoimmune diseases, as well as in tumour progression [9]. The Monocyte Chemoattractant Protein-1 (MCP-1) and Macrophage inflammatory Protein (MIP-1α) play a significant role in the development of coronary artery disease [12]. Several cardiovascular diseases are associated with high levels of pro-inflammatory cytokines that promote the formation of atherosclerotic plaque through an accumulation of macrophages, lipid-laden cells, T cells, and other kinds of degenerative matter in the inner layer of blood vessel walls. Subsequently, several inflammatory molecules are released by the action of macrophages and lead to tissue damage and consequently to inflammation [12]. MCP-1 is produced by different types of cells (monocytes, macrophages, smooth muscle cells, endothelial cells, etc.); MIP-1α is induced by several pro-inflammatory agents and is negatively affected by anti-inflammatory agents [13]. Both are associated with the risk of developing inflammation, cardiovascular disease (specifically atherosclerosis), increased risk of myocardial infarction, and death [14]. 

According to the literature, physical activity is a very important tool for cardiovascular health. At the hemodynamic and cardiovascular level, physical exercise, mainly exercise with aerobic characteristics, can contribute to the reduction of the luminal diameter of the arteries, since it has a localized effect and/or reflects the increase in blood flow [15]. In addition, aerobic exercise is also considered a first-line strategy to prevent and treat hypertension [8]. Hypertensive individuals are encouraged to participate in aerobic exercise programs aimed at reducing systolic and diastolic blood pressure [16]. This exercise-induced reduction is also accompanied by improvements in arterial and peripheral hemodynamic factors [16]. Combined physical exercise programs (aerobic and muscle strength exercise) have also been shown to be effective in improving hemodynamic and cardiovascular parameters in the older population. A study by Son et al. [17] found that a 12-week combined exercise program provided significant changes in arterial stiffness, as well as in systolic and diastolic blood pressure, in elderly women (75 ± 2 years). However, the meta-analysis carried out by Montero et al. [18] concluded that the combined physical exercise benefits do not appear to differ significantly when compared to aerobic exercise alone.

Regular physical exercise can cause a reduction around −3.5 mmHg in systolic blood pressure and −3 mmHg in diastolic blood pressure [19]. Physical exercise in an aquatic environment, due to immersion, promotes physiological adjustments that can beneficially affect blood pressure, namely by reducing sympathetic activity and redistributing blood volume from the lower limbs and the abdominal area to the upper part of the body, thus causing a reduction in peripheral vascular resistance [19].

Regarding the biochemical markers related to cardiovascular diseases, studies have shown that physical exercise is also an effective tool to improve these factors. In Martins et al. [20], two exercise programs (aerobic and muscle strength exercise) were tested in a group of 63 sedentary elderly participants (76.0 ± 8.0 years), and both interventions produced beneficial effects on their lipidic profile (TG, TC, LDL, and HDL). According to Gleeson [21], physical exercise can be a way of inhibiting the release of chemokines in human adipose tissue, contributing to the reduction of the pathogenesis of various cardiovascular diseases.

Physical exercise has shown evidence of being an effective way to improve cardiovascular markers, but there are few studies that test its effectiveness in aquatic environments [7]. The exercise in aquatic environment, compared to exercise on land, provides specific mechanical advantages due to the principles of buoyancy, viscosity, and drag. These advantages make the practices of exercise more pleasant and safer for the joints among the older population [22]. For this reason, the purpose of this study is to test the impact of different physical exercise programs in aquatic environments (continuous aerobic program, aerobic interval program, and combined program) on the IMT, hemodynamic parameters, and biochemical markers associated with cardiovascular diseases in community dwelling older people. Considering the low number of studies published that combine imaging technology, biochemical markers, and aquatic exercise interventions [7,18,20], we believe that this study will provide relevant information on the effect of these three exercise programs on cardiovascular risk indicators.

## 2. Materials and Methods

### 2.1. Study Design

A randomized controlled trial was conducted in Beira Interior Region, Portugal. A sample of non-institutionalized older participants were submitted to a 28-week aquatic exercise intervention. The entire study protocol was previously published by Ferreira et al. [23]. Intima-media thickness of carotid arteries (IMT), blood pressure, lipid blood profile, and chemokines blood concentration (MCP-1 and MIP-1α) were measured in all participants, undergoing three different physical aquatic exercise programs: continuous aerobic (AerG), interval aerobic exercise (IntG), and combined exercise (ComG). A fourth group of participants was also selected as the control group (CG). Data were collected at two specific time moments, namely pre-intervention (baseline, M1) and post-intervention (after 28 weeks, M2). This study was carried out according to the recommendations of the Declaration of Helsinki for Human Studies. The protocol was approved by the Ethics Committee for Health of the Faculty of Sport Sciences and Physical Education, University of Coimbra (reference: CE/FCDEF-UC/00462019). Written informed consent was obtained from all participants prior to any protocol-specific procedures.

### 2.2. Participants and Sample Size

The size and statistical power of the sample were calculated using the G*Power software application (University of Dusseldorf, Dusseldorf, Germany) [24]. The following parameters were considered: F test (ANOVA); effect size, 0.25; α-level, 0.05; statistical power, 0.95; number of groups, 4; number of measures, 2 (pre and post intervention); margin of possible losses and refusals, 30%. Therefore, the initial size of the total sample was estimated at 76 participants.

Initially, 174 individuals from the community were invited to participate in the study. After the application of the inclusion and exclusion criteria, 152 individuals were randomized into the four groups mentioned above: AerG, *n* = 36; IntG, *n* = 41; ComG, *n* = 48 and CG, *n* = 27. According to the experience of the research team and previous studies, the dropout rate from exercise programs among elderly populations is high, so we gathered more participants to fulfil the sample size and compensate for potential dropouts [25,26,27]. A simple randomization method was used. An external investigator used a computer-generated list of random numbers to allocate participants to each group. The investigators were blinded for the randomization of the groups.

The following inclusion criteria were applied: (a) participants from both sexes; (b) 65 years or older; (c) non-institutionalized; (d) autonomy to travel from their residence to Sertã municipal swimming pool; (e) filling out the informed consent form; (f) individuals with medical authorization to practice physical exercise in an aquatic environment, in cases they had some type of clinical condition or comorbidity. The following exclusion criteria were also defined: (a) individuals with clinically diagnosed pathologies putting their and others’ health at risk while doing physical exercise in an aquatic environment; (b) individuals that obtained a score of less than 9 points in the Mini-Mental States Examination (indicating severe cognitive impairment) or were clinically diagnosed with a mental illness; (c) an attendance of less than 50% to physical exercise sessions; (d) participants who failed to complete all proposed assessment tests.

The AerG, IntG, and ComG groups performed physical exercise in an aquatic environment during a period of 28 weeks. Participants from the CG group were asked to maintain their normal daily activities without performing any type of systematic physical exercise during the same time period. The participants in the control group were age-matched persons who did not perform any type of regular systematic exercise.

Fifty participants dropped out from the study for the following reasons: personal reasons (*n* = 11); less than 50% of attendance of the exercise sessions (*n* = 14); not completing all the assessment tests (*n* = 12); injury not related with the exercise intervention (*n* = 4); disease (*n* = 9). Consequently, 102 participants completed the entire process (AerG: *n* = 25, 71.44 ± 4.84 years old, women—80%, taking regular medication—92%, with cardiometabolic diseases—76%; IntG: *n* = 28, 72.64 ± 5.22 years old, women—89.3%, taking regular medication—96%, with cardiometabolic diseases—82%; ComG: *n* = 29, 71.90 ± 5.67 years old, women—75.9%, taking regular medication—90%, with cardiometabolic diseases—79%; CG: *n* = 20, 73.60 ± 5.25 years old, women—55%, taking regular medication—95%, with cardiometabolic diseases—80%). The cardiometabolic diseases included diabetes, hypertension, and hypercholesterolemia. Figure 1 shows the entire allocation process for the different groups.

### 2.3. Outcomes Measurements

#### 2.3.1. Sample Characterization

To characterize the sample, values related to anthropometry and physical fitness were collected. Height (Hgt) was assessed using a portable Seca Bodymeter^®^ stadiometer (model 208, Hamburg, Germany) with an accuracy of 0.1 cm. Weight (Wgt), body mass index (BMI), visceral fat (VF), fat mass (FM), and lean body mass (LBM) were evaluated using the TANITA BC-601 impedance scale (Tokyo, Japan). Functional fitness was assessed using the following tests from the Senior Fitness Test set, developed by Rickli and Jones [28] and validated for the Portuguese population by Baptista et al. [29]: muscle strength of the lower (MI) and upper (MS) limbs, with the Chair Stand test (30 s—CS) and Arm Curl test (30 s—AC), respectively (repetitions/30 s); aerobic capacity, with the Two-Minute Step test (2 m—ST) (number of steps); the flexibility of MI and MS, with the Chair Sit and Reach test (CSR) and Back Scratch test (BS), respectively (centimetres); agility and dynamic balance, with the Timed Up and Go test (TUG) (seconds). The handgrip strength was also evaluated, with the Hang Grip test (HG), using the Jamar hand dynamometer (Lafayette Instrument Company, Lafayette, IN, USA) (kg).

#### 2.3.2. IMT and Hemodynamic Parameters

The intima-media thickness of the carotid arteries (IMT) was evaluated with a General Electric (GE^®^) portable ultrasound machine, VIVID model, with a 9 L linear probe (4 to 12 MHz). All examinations were performed by a highly trained technician. The measurements and data reanalysis were interpreted by two independent technicians, with all data being recorded in digital support. All measurements were performed in the space between 10 and 20 mm before the carotid bifurcation (carotid bulb), allowing the measurement of the IMT in the most distal wall, which is the one with the best definition.

The participants performed the examination in a temperature-controlled room (22 °C to 24 °C), lying on an examination table, in a supine position, with their heads turned to the side (45°). With the use of the real-time B-mode Echo Doppler ultrasound technique, we obtained an image of the carotid artery in the longitudinal plane and transverse plane in order to confirm the IMT measurements.

The images were also used to obtain the systolic diameters (SD) and diastolic diameters (DD). The peak systolic velocity (PSV) and the endo-diastolic velocity (EDV) were measured with the Pulse-Doppler technique in association with the continuous Doppler. For higher accuracy in the velocity calculations, the insonation angle was set at 60°. This allowed us to obtain correct values with respect to the Doppler equation.

The measurements of heart rate (HR), systolic blood pressure (SBP), and diastolic blood pressure (DBP) were performed with the use of a digital automatic sphygmomanometer from Riester (Model Ri-ChampionN^®^, Jungingen, Germany). All the measurements were taken after the participants were at rest for a period of 5 min, sitting and silent, in a room with controlled temperature (22 °C to 24 °C). After this period, participants maintained their comfortable sitting position in a chair, keeping their torsos straight, and the right upper limb stretched and placed on top of a table. Then, a cuff was placed on their right upper limbs, aligned with the brachial artery, at the level of the heart, and adjusted to the perimeter of the arm. Three measurements were taken, with intervals of 1–2 min between them, to check what the average blood pressure was. The mean of the three measurements was considered according to the Guidelines for the Management of Arterial Hypertension [30] of the European Society of Hypertension (ESH) and the European Society of Cardiology (ESC).

#### 2.3.3. Biochemical Markers

Fasting blood samples (15 mL) were collected from the ante cubital vein by a registered nurse. The blood sample was used by the clinic to assess lipidic panel values: cholesterol total (TC), high-density lipoprotein (HDL), low-density lipoprotein (LDL), triglycerides (TG), and glucose (GLU). The atherogenic index (AI) was calculated using the TC/HDL ratio and considered normal for values less than five. Next, the test tubes were centrifuged for 10 min at 3500 rpm, and plasma and serum were retrieved and stored in cryogenic test tubes at −80 °C until further use. Levels of Monocyte Chemoattractant Protein-1 (MCP-1/CCL-2) and Macrophage Inflammatory Protein-1 alfa (MIP-1α/CCL3) were subsequently analysed by ELISA (Invitrogen^®^, Alfagene, Portugal) according to the manufacturer’s instructions.

### 2.4. Intervention Protocol

The exercise programs were implemented by sport science and fitness experts, with specific training in hydrogymnastics and developed according to the exercise prescription guidelines recommended by the American College of Sport Medicine (ACSM) for the elderly [31].

All exercise programs sessions had a duration of 45 min, twice a week, for 28 weeks and were performed in a water environment (the water level was between 0.80 and 1.20 m with a temperature of approximately 32 °C), using musical rhythm as a tool to control the intensity of the exercise. Sequences of aquatic exercises, previously defined and selected according to the objectives of each program, were applied. Water exercise sessions were organized into three different parts: the initial, main, and final part.

The initial part or warm-up lasted between 10 and 15 min, at low intensity (30–40% max HR), and was the same in the three water exercise programs. During this initial part, it is intended that participants will adapt to the aquatic environment, i.e., to the water temperature, and provide muscular and metabolic stimulations to prepare the body for the main part of the session. Thus, simple exercises in water were used, such as displacements and isolated movements, with a progressive increase in complexity and intensity throughout the initial part.

The main part for each one of the three water exercise program sessions had a duration of 20 to 30 min and had the following characteristics (Figure 2):-Continuous aerobic (AerG): exercise aerobic (weeks 1–13, 60–65% maximum HR; weeks 14–28, 65–70% HR);-Interval aerobic (IntG): exercise aerobic different intensities (weeks 1–13, 60–65% maximum HR interval to 70–75% maximum HR, weeks 14–28, 65–70% maximum HR interval to 75–80% maximum HR). In IntG, the main part of the sessions consisted of several series (1’ recovery interval with 30’’ with higher intensity);-Combined (ComG): exercise aerobic (weeks 1–13, 60–65% maximum HR; weeks 14–28, 65–70% maximum HR) and muscle strength (weeks 1–13, 2 sets of 12 repetitions; weeks 14–28, 3 sets of 16 repetitions; 6–7 pois in the Borg Scale).

The final part of the water exercise sessions lasted between 5 and 10 min and was the same for each of the three water exercise programs. This part consisted of two phases: return to calm, where relaxation exercises were applied to bring back the participants’ heart rate to values close to the resting state, and stretching exercises stimulating a greater range of motion, used to stretch the main muscle groups used throughout the sessions.

### 2.5. Monitoring the Intensity of the Exercise of Programs

For safety and intensity target control reasons, all participants randomly used heart rate monitors (Polar, R800CX) (Polar Electro Oy, Professorintie 5, FI-90440 Kempele, Finland) during the exercise sessions, in all three water exercise programs. Depending on the heart rate values obtained, intensity adjustments to the training plan were performed to maintain the intensity target defined for each water exercise program.

For safety and intensity target control reasons, the intensity of the different water exercise programs was predicted indirectly using Karvonen’s Formula [32]:Target heart rate = ((maximal heart rate − resting heart rate) × % intensity) + resting heart rate

Additionally, and to calculate the maximal heart rate, we used the [33] following formula for the elderly:Maximal HR = 207 beats per minute − (0.7 × chronological age)

### 2.6. Statistical Analysis

The collected data were subjected to descriptive statistical analysis where values such as maximum, minimum, mean, and standard deviation were calculated for each variable in each assessment moment. Afterwards, data normality was tested by considering the response to three conditions: z-values from the skewness and kurtosis tests; *p*-values from the Shapiro–Wilk test; and visual inspection of generated histograms. All longitudinal comparisons were performed using complete case analysis. Parametric data were analysed using the Student’s t-test for independent samples to compare the different moments (M1 and M2) and the one-factor ANOVA test and post hoc Tukey’s test to analyse the differences between groups. Nonparametric data were analysed using the Wilcoxon test to compare the different moments (M1 and M2), and the Kruskal–Wallis and Bonferroni tests were used to analyse differences between groups. Associations between variables were analysed using Pearson’s correlation coefficient values and interpreted as follows [34]: r = 0.10 to 0.29 means weak correlation; r = 0.30 to 0.49 means moderate correlation; and r = 0.50 to 1.0 means strong correlation. Statistical analysis was performed using the Statistical Package for the Social Sciences (SPSS) statistical software, version 27.0 (IBM, New Orchard Road Armonk, New York, United States). The level of significance used was *p* ≤ 0.05.

## 3. Results

The base line values characterizing of each group (mean and standard deviation) are shown in Table 1. No significant statistical differences were found between the groups before the intervention (M1), which suggests that all study groups had similar characteristics regarding anthropometry and functional fitness (*p* > 0.05). Exercise frequency rate was 75.4% in AerG, 69.3% in IntG and 77.7% in ComG.

Among the different groups of evaluated variables, intima-media thickness of the carotid arteries, blood pressure, lipid profile, fasting glucose, and MCP-1 and MIP-1 markers were analysed for correlations, in the two evaluation phases (M1 and M2) (see Figure 3). Weak and positive correlations were found between SBP and SD-L (r = 0.28) and DD-R (r = 0.25). Contrastingly, we found moderate and negative correlations between HDL and SD-L (r = −0.30) and weak and negative correlations between HDL and DD-R (r = −0.20). There were no statistically significant correlations between the fasting glucose, MCP-1 and MIP-1 markers, and the remaining variables analysed.

The variation in the results of the hemodynamic and cardiovascular parameters, which were analysed by type of aquatic exercise program, before and after the intervention, are shown in Table 2. An example of an image of the carotid artery of a person who participated in the two evaluation phases (M1 and M2) is presented in Figure 4. Although there were no statistically significant differences in IMT after intervention in the different exercise groups, significant reductions were observed in some participants of the exercise groups. This reduction can be explained by the high adherence to the physical exercise program, which on average was 85.6%.

The general results of the hemodynamic parameters showed no significant differences between the groups before and after the intervention (*p* > 0.05). As a result of the intervention with the aquatic exercise program, significant differences were found between M1 and M2 in AerG for DD-R (*p* = 0.039; Δ = −4.5%), in IntG for PSV-R (*p* = 0.024; Δ = −11.3%), and in ComG for DD-L (*p* = 0.037; Δ = −4.8%), for EDV-L (*p* = 0.027; Δ = −10.8%), and DD-R (*p* = 0.029; Δ = −4.8%). In the CG, there was a significant increase for EDV-L (*p* = 0.003; Δ = 13.8%), while a significant reduction in the same variable was observed in ComG. In regard to the variables IMT-L, SD-L, PSV-L, IMT-R, SD-L, and EDV-R, no significant changes were observed after the intervention (*p* > 0.05).

Regarding the blood pressure parameters results, no significant differences were found between the groups before and after the intervention (*p* > 0.05). As a result of the intervention with the aquatic exercise programs, significant differences were found in AerG for SBP (*p* = 0.013; Δ = −4.5%) and DBP (*p* = 0.002; Δ = −5.2%), in IntG for DBP (*p* = 0.046; Δ = −3.9%), and in ComG for SBP (*p* = 0.018; Δ = −3.6%), DBP (*p* = 0.004; Δ = −5.1%) and HR (*p* = 0.010; Δ = −5.4%). In the control group, there were no significant changes (*p* > 0.05).

The variation in the results for the cardiovascular risk biomarkers, analysed by type of aquatic exercise program, before and after the intervention, are presented in Table 3. 

The overall lipid profile results revealed that no significant differences were found between the groups before the intervention (*p* > 0.05). After the intervention, significant differences were found for AI between IntG and CG (*p* = 0.008) and between ComG and CG (*p* = 0.009). In both cases, the CG obtained a less satisfactory result. As a result of the intervention with the aquatic exercise program, significant differences were found in AerG (*p* = 0.006; Δ = −5.3%) and IntG (*p* = 0.041; Δ = −5.3%) for GLU, and in CG for AI (*p* = 0.003; Δ = 14.3%). In regard to the variables TC, HDL, LDL, and TG, there were no significant changes (*p* > 0.05). As for the chemokines MCP-1 and MIP-1α, statistically significant differences were found between the groups, before and after the intervention (*p* < 0.05). From M1 to M2, statistically significant differences were found in AerG for MCP-1 (*p* = 0.001; Δ = −41.8%) and MIP-1α (*p* = 0.009; Δ = −2.2%) and in ComG for MCP-1 (*p* = 0.033; Δ = −16.5%). For the IntG and CG, no statistically significant differences were found in both chemokines (*p* > 0.05).

## 4. Discussion

The purpose of this study was to test the impact of different aquatic exercise programs (continuous aerobic exercise, aerobic exercise interval, and combined exercise) on the IMT and hemodynamic and biomarkers of cardiovascular diseases in community dwelling older people. Preliminary systematic research revealed the innovative characteristics of the present study, since there were very few studies that have evaluated similar variables with aquatic exercise interventions in older adults.

Although no changes were found in IMT or in the variables related to the hemodynamic parameters, significant reductions were observed for DD-R in AerG and ComG, for PSV-R in IntG, and for DD-L in ComG. In EDV-L, there was a significant reduction in ComG, while in CG a significant increase was observed.

A study by Santos [7] reported the effectiveness of a 10-month continuous aerobic aquatic exercise program in older women (64.05 ± 5.91 years), who were divided into two groups: older women who started practising exercise in an aquatic environment (continuous aerobic program), for the first time (beginners) and elderly women who had already practised physical exercise in an aquatic environment (continuous aerobic program) during the previous school year (trained). The results showed that in both groups there was a significant reduction in IMT, with a magnitude effect of 36.4% and that exercise had a very strong effect on the change in IMT. Regarding SD and DD, there was a significant reduction in both groups. Additionally, a reduction in PSV and EDV was reported in both groups. Such a reduction was significant in the trained group. Similarly, in the study by Park and Park [15], the effectiveness of a combined exercise program (aerobic exercise and muscle strength exercise) was tested in a group of older women (65–77 years) for 24 weeks. The results showed that these types of programs can lead to a significant reduction in IMT. In another study [35], the impact of two exercise programs (high-intensity interval exercise and continuous aerobic exercise) was tested, and the results showed that, although they did not lead to significant statistical differences, both programs resulted in a reduction of the IMT.

The results from the present study are in line with those in the existing literature. In IMT-R, although there were no significant statistical differences, a trend towards a reduction was visible in the three exercising groups, whereas in IMT-L only a reduction in IntG was visible, with the value of the mean remaining the same in the other groups. As reported by Santos [7], our results also showed a reduction in SD, DD, PSV, and EDV in all exercise groups. This reduction was significant for DD-L in ComG, for DD-R in AerG and ComG (in regard to the latter, ComG recorded a higher variation of the mean), for EDV-L in ComG, and for PSV-R in IntG. In the CG, a significant increase in EDV-L was evident. According to Homma et al. [36], there is a strong and positive correlation between advancing age and an increase in IMT (r = 0.83). In our study, although the results did not significantly decrease in all variables, in the exercise groups, they did not increase either. For this reason, we can say that all exercise programs can be considered as factors of hemodynamic balance and thus contribute to the reduction in stress on the arterial wall and reduce the risk of developing cardiovascular diseases. The combined program seems to be slightly more beneficial compared to the other ones, due to the fact that it resulted in significant differences in a higher number of variables. Additionally, compared to the study by Santos [7], our intervention lasted 7 months, while in Santos’s (2020) study, the intervention lasted 10 months. These data suggest that a 7-month intervention with aquatic exercise may be sufficient to cause changes in these hemodynamic variables.

Significant reductions were also found for SBP in the AerG and ComG, as well as in the three-exercising groups for DBP, and for HR in the ComG. 

A study by Park et al. [37] tested the effectiveness of a water walking program on blood pressure values in a group of women (70 ± 10.0 years), and the results showed that, after the intervention, the exercise group had a significant reduction in HR values. The SBP and DBP values were also reduced, although not significantly. In land environments, a study by Son et al. [17] reported significant reductions in SBP and DBP after intervention with a combined exercise program (muscle strength exercise and aerobic exercise) in a group of hypertensive women (75 ± 2 years old). The same results were confirmed by Park et al. [38] using the same type of program (combined exercise), which led to a decrease in SBP and DBP values. This decrease was statistically significant in SBP.

In our study, the three exercise groups showed reductions in SBP, DBP, and HR. The aerobic exercise and combined exercise program seem to be the most beneficial. Both showed similar results in SBP and DBP, with the aerobic exercise program leading to a slightly higher decrease in the mean compared to the combined exercise program. In HR, the combined exercise program was the only one where the decrease was statistically significant. We may conclude that the combined exercise program may be more beneficial for cardiovascular parameters, as in addition to having showed similar results to the aerobic program in SBP and DBP, it also led to a significant decrease in HR.

In regard to the biochemical markers, significant reductions were noticed for GLU in AerG and IntG, and a significant increase was noticed for the AI in CG (14.3%). After the intervention, significant differences were found in the AI between IntG and CG and between ComG and CG.

In a study with participants with type 2 diabetes [39], after an intervention in an aquatic environment with a moderate aerobic exercise program, the results showed a significant decrease in GLU, TC, LDL, and TG values. Another study [40] tested the effectiveness of two different aquatic exercise programs: aerobic exercise and muscle strength exercise. Both programs led to a significant decrease in the TC, LDL, and TG values and a significant increase in HDL values. Therefore, they concluded that the two types of aquatic exercise programs had identical benefits, and both could contribute to a reduction in the risk of cardiovascular events. Similar results were reported on a land study [41], where the effectiveness of two exercise programs (high-intensity interval exercise and moderate-intensity continuous aerobic exercise) was compared, with the results showing a decrease in TC, LDL, and TG values and an increase in the HDL values. In comparison with the aerobic exercise program, the interval exercise program showed results that were more beneficial.

As in the study of Cugusi et al. [39], the present study noticed a decrease in the GLU values in the three exercise groups, although only statistically significant in AerG and IntG. Our results showed that AerG and IntG are equally beneficial in reducing GLU. Regarding TC, HDL, LDL, and TG, the results did not show significant changes, and they remained within the reference values. Therefore, aquatic physical exercise may contribute to the balance of the lipid profile.

In the plasma chemokine levels, significant reductions were noticed for MCP-1 and MIP-1α in AerG, and for MCP-1 in ComG.

Among long-term intervention studies, the one conducted by Kraemer et al. [42], which consisted of carrying out a 12-week muscle strength training program with and without supplementation in young adults, MCP-1 values increased after the intervention, while the MIP-1α values decreased, regardless of the supplementation. In the study of Barry et al. [43], who tested the impact of two physical exercise programs (high-intensity interval exercise and moderate-intensity continuous exercise) in obese adults, the results showed that, although no statistically significant differences were found, the MCP-1 values were reduced in both groups after the intervention, and that the MIP-1α values increased after the intervention of the high-intensity interval program and decreased after the intervention of the moderate-intensity continuous program.

In our study, despite the fact that the intervention took place in a different context, the results partly reached the same conclusions. As in the Barry et al. [43], our results reinforce the idea that long-term continuous aerobic exercise reduces the values of MCP-1 and MIP-1α. The results of the interval aerobic program showed a reduction in MCP-1 and an increase in MIP-1α. It is possible that higher intensity exercise levels may lead to increased MIP-1α levels as a response to muscle micro-injuries. In both cases, the changes were not significant, just like in the Barry et al.’s study [43]. Finally, in the combined exercise program (continuous aerobic exercise and muscle strength exercise), the results showed a reduction in both chemokines.

MCP-1 appears to be present in the initial phase of atherosclerotic lesion formation, i.e., in the thickening of the tunica intima and fatty streaks, which suggests that this chemokine may contribute to an early influx of monocytes into blood vessels [13]. In the intimal layer of the carotid arteries, chemokines recruit monocytes that trigger the development of foam cells, leading to intimal erosion, which is caused by atherosclerosis and consequently leads to ischemia [12]. The results found in the present study suggest that aquatic physical exercise, especially exercise with aerobic characteristics, helps to prevent the development of cardiovascular diseases by contributing to the reduction in chemokines MCP-1 and MIP-1α.

Exercise in aquatic environments, due to the specific proprieties of water that make the exercise practice safer and more comfortable for the elderly population, reduces the risk of traumatic fractures and joints by giving them less stress and lower impact [44]. The viscosity of the water provides a comfortable resistance that allows the development of muscle strength [22]. The hydrostatic pressure, the force of the viscosity, and the turbulence created by the water during the physical exercise sessions provide a different proprioceptive and sensorial feedback from the land environment and provide periods of instability that must be overcome. These aspects benefit the postural control and balance system [45,46]. The practice of exercise in this environment seems to be a viable strategy, because in addition to its specific characteristics and added benefits, it seems to be equally effective for cardiovascular health when compared to exercise on land.

Study limitations included the fact that a simple randomization method was used, instead of using a blocked randomization method, which would guarantee a balance in the number of participants in each group, reducing sequence unpredictability. Another limitation was the fact that the level of physical activity of the participants in the control group was not evaluated. Although participants were not involved in regular systematic exercise, this assessment would have contributed to a stronger study. The limited number of publications with interventions in an aquatic environment in older populations was also a limitation for the discussion of results.

More intervention exercise studies in aquatic environments are needed in order to reinforce the results found in the present study and better understand the effects and effectiveness of aquatic exercise programs on markers of cardiovascular health.

## 5. Conclusions

Aquatic physical exercise, regardless of the type of program, seems to lead to benefits in cardiovascular variables, and this type of intervention may be a viable alternative when land-based exercise is not possible and/or desired.

In relation to the intima and average thickness of the carotid arteries, we can say that all the aquatic exercise programs studied (i.e., the continuous aerobic program, the aerobic interval program, and the combined program) can be considered as factors for hemodynamic balance and thus contribute to the reduction of stress on the arterial wall and reduce the risk of cardiovascular diseases. The combined program seems to be slightly more beneficial, compared to the other programs due to the fact that it resulted in significant differences in a higher number of variables.

As for blood pressure, the three exercise groups showed reductions in SBP, DBP, and HR. However, the aerobic and combined exercise programs proved to be more beneficial for blood pressure and present statistically significant differences for both variables (with the same mean variation). Additionally, the combined program also showed a statistically significant reduction for HR.

Regarding the metabolic profile, our results showed that AerG and IntG are equally beneficial in reducing GLU. Regarding TC, HDL, LDL, and TG, the results did not show significant changes; however, they remain within the reference values. We concluded that aquatic physical exercise may contribute to the balance of the lipid profile.

Finally, as for chemokines, the results suggest that aquatic physical exercise, especially that with aerobic characteristics, helps prevent the development of cardiovascular diseases by contributing to the reduction in chemokines MCP-1 and MIP-1α and the subsequent infiltration of monocytes and formation of atherosclerotic plaques.

## Figures and Tables

**Figure 1 ijerph-19-03377-f001:**
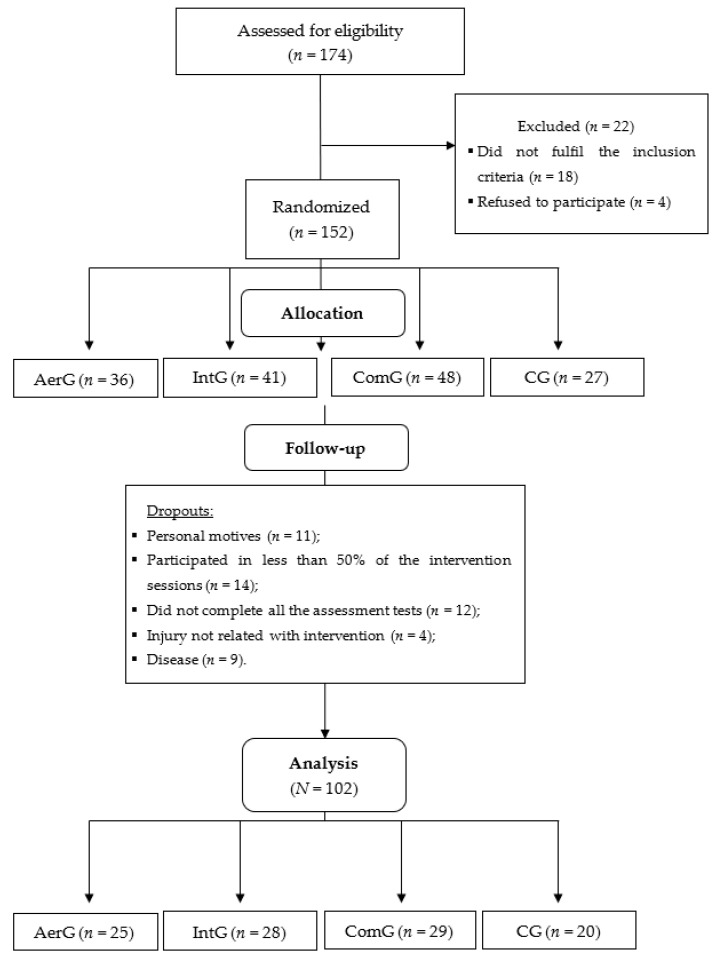
Allocation process for the different groups: Continuous aerobic group (AerG); Aerobic interval group (IntG); Combined group (ComG); Control group (CG).

**Figure 2 ijerph-19-03377-f002:**
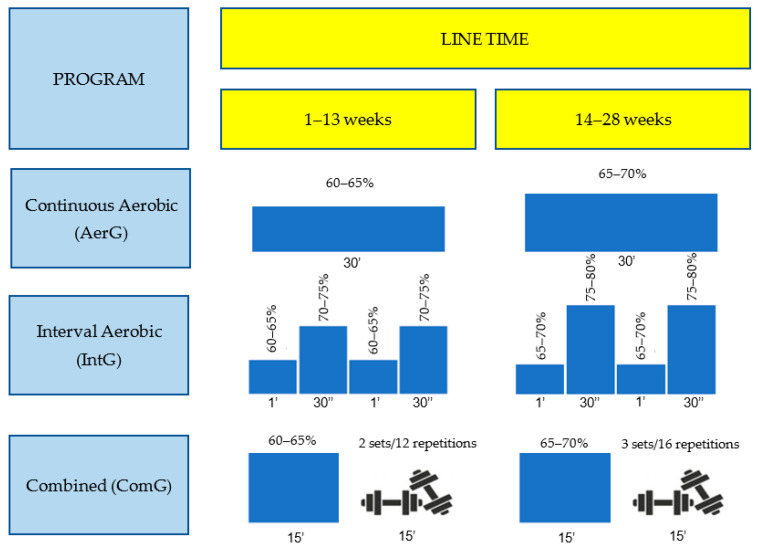
Characteristics of the main part for the three physical exercise aquatic programs (continuous aerobic, interval aerobic, and combined).

**Figure 3 ijerph-19-03377-f003:**
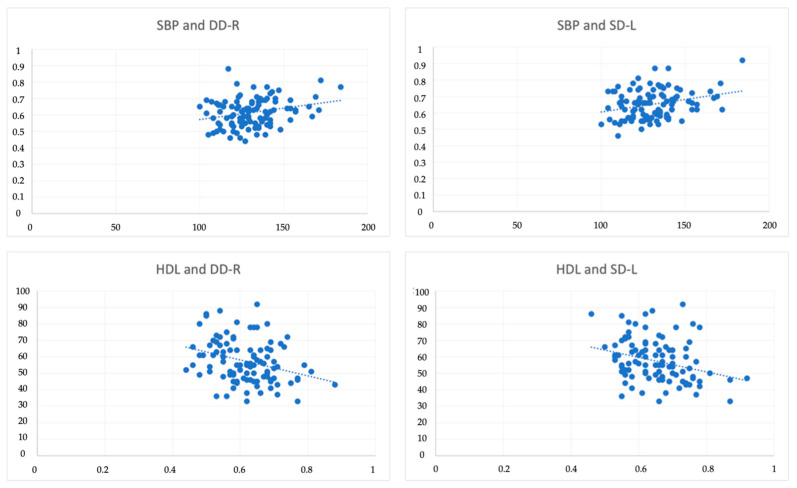
Relationships between SBP-DD-R and SBP-SD-L, and HDL-DD-R and HLD-SD-L. Notes: systolic blood pressure (SBP), diastolic diameter—right (DD-R), systolic diameter—left (SD-L), high-density lipoprotein (HDL).

**Figure 4 ijerph-19-03377-f004:**
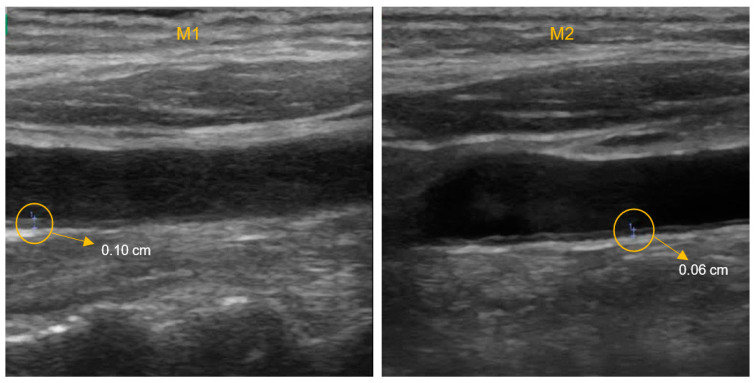
Images of the carotid artery of a study participant at different times of assessment: before the intervention (M1), the participant had an intima and media thickness of the carotid artery (IMT) of 0.10 cm; after the intervention (M2), a reduction in the IMT to 0.06 cm was verified.

**Table 1 ijerph-19-03377-t001:** Characterization of the baseline sample (M1).

Characteristic	AerG (*n* = 25)	IntG (*n* = 28)	ComG (*n* = 29)	CG (*n* = 20)	*p* Value
	Mean (SD)	Mean (SD)	Mean (SD)	Mean (SD)	
AG (years)	71.44 (4.8)	72.64 (5.2)	71.90 (5.7)	73.60 (5.3)	0.504
Hgt (m)	1.58 (0.7)	1.56 (0.7)	1.57 (0.7)	1.60 (0.9)	0.331
WGT (kg)	70.5 (8.1)	71.3 (14.3)	75.1 (11.0)	75.5 (13.3)	0.334
BMI (kg/m^2^)	28.20 (3.3)	29.10 (4.8)	30.80 (5.3)	29.50 (5.8)	0.272
VF (%)	11.0 (3.0)	12.0 (3.0)	13.0 (3.0)	13.0 (6.0)	0.128
FM (%)	38.9 (7.3)	41.0 (6.7)	40.3 (9.8)	34.9 (10.9)	0.134
LBM (%)	26.5 (4.3)	24.5 (3.0)	25.5 (4.3)	27.7 (4.7)	0.079
2 m-ST (no of steps)	80.9 (17.4)	71.5 (16.5)	81.6 (19.2)	74.3 (18.9)	0.069
CSR-R (cm)	−0.5 (6.6)	−3.7 (10.6)	−3.5 (7.8)	−7.6 (9.7)	0.099
CSR-L (cm)	0.6 (7.2)	−3.9 (9.9)	−5.8 (9.9)	−3.5 (7.3)	0.054
BS-R (cm)	−9.9 (10.4)	−11.9 (11.5)	−14.3 (9.7)	−16.6 (9.9)	0.157
BS-L (cm)	−14.4 (7.2)	−17.4 (8.6)	−21.0 (10.8)	−20.6 (10.7)	0.056
TUG (s)	6.1 (1.1)	7.4 (1.8)	7.4 (3.0)	6.8 (1.7)	0.110
30 s-CS (reps/30 s)	15.0 (3.0)	13.0 (4.0)	13.0 (3.0)	15.0 (5.0)	0.185
30 s-AC (reps/30 s)	21.0 (6.0)	17.0 (7.0)	20.0 (5.0)	19.0 (6.0)	0.119
HG-R (kg)	22.0 (6.0)	21.0 (9.0)	21.0 (9.0)	24.0 (9.0)	0.411
HG-L (kg)	21.0 (6.0)	20.0 (9.0)	21.0 (9.0)	21.0 (10.0)	0.578

Note: Age (AG); height (Hgt); weight (Wgt); body mass index (BMI); visceral fat (VF); fat mass (FM); lean body mass (LBM); two-minute step test (2 m-ST); chair sit and reach test—right (CSR-R); chair sit and reach test—left (CSR-L); back scratch test—right (BS-R); back scratch test—left (BS-L); timed up and go test (TUG); chair stand test (30 s-CS); hand grip test—right (HG-R); hand grip test—left (HG-L).

**Table 2 ijerph-19-03377-t002:** IMT and Hemodynamic parameters results.

	AerG			IntG			ComG			CG			Time × Group (M1)	Time × Group (M2)
	M1 Mean (SD)	M2 Mean (SD)	Time (*p*)	M1 Mean (SD)	M2 Mean (SD)	Time (*p*)	M1 Mean (SD)	M2 Mean (SD)	Time (*p*)	M2 Mean (SD)	M1 Mean (SD)	Time (*p*)		
IMT-L (cm)	0.07 (0.02)	0.07 (0.02)	0.293	0.08 (0.02)	0.07 (0.02)	0.376	0.07 (0.02)	0.07 (0.02)	0.707	0.07 (0.01)	0.07 (0.02)	0.979	0.989	0.501
SD-L (cm)	0.69 (0.12)	0.67 (0.09)	0.177	0.65 (0.1)	0.65 (0.09)	0.539	0.66 (0.1)	0.64 (0.09)	0.105	0.65 (0.06)	0.66 (0.06)	0.088	0.605	0.546
DD-L (cm)	0.64 (0.12)	0.62 (0.09)	0.423	0.61 (0.09)	0.59 (0.09)	0.105	0.63 (0.09)	0.60 (0.09)	0.037 **	0.61 (0.06)	0.62 (0.06)	0.558	0.640	0.264
PSV-L (cm/s)	76.79 (21.45)	80.31 (19.22)	0.242	82.51 (19.78)	77.21 (16.26)	0.104	79.38 (18.77)	78.73 (17.06)	0.315	77.28 (15.91)	80.24 (17.9)	0.097	0.508	0.835
EDV-L (cm/s)	19.28 (6.24)	19.95 (5.8)	0.397	20.8 (5.43)	20.26 (4.93)	0.581	22.38 (4.78)	19.97 (6.56)	0.027 *	19.7 (5.32)	22.41 (5.62)	0.003 *	0.090	0.443
IMT-R (cm)	0.08 (0.01)	0.07 (0.01)	0.808	0.08 (0.02)	0.07 (0.02)	0.055	0.08 (0.02)	0.07 (0.01)	0.205	0.08 (0.02)	0.08 (0.02)	0.308	0.654	0.698
SD-R (cm)	0.70 (0.12)	0.68 (0.09)	0.178	0.66 (0.09)	0.65 (0.09)	0.381	0.67 (0.08)	0.66 (0.09)	0.249	0.68 (0.08)	0.66 (0.08)	0.353	0.519	0.758
DD-R (cm)	0.66 (0.12)	0.63 (0.09)	0.039 *	0.62 (0.08)	0.61 (0.09)	0.282	0.63 (0.09)	0.60 (0.09)	0.029 *	0.62 (0.07)	0.63 (0.07)	0.813	0.316	0.543
PSV-R (cm/s)	74.6 (23.04)	74.1 (17.56)	0.904	74.63 (18)	66.23 (17.68)	0.024 *	81.23 (15.74)	77.25 (12.81)	0.087	74.27 (16.9)	73.11 (16.31)	0.763	0.136	0.106
EDV-R (cm/s)	19.55 (7.29)	18.89 (5.41)	0.716	19.39 (5.44)	18.26 (4.8)	0.186	20.74 (5.29)	19.18 (6.34)	0.064	18.43 (4.63)	19.27 (5.35)	0.432	0.169	0.910
SBP (mmHg)	134 (13.0)	128 (16.0)	0.013 *	135 (12.0)	132 (13.0)	0.184	138 (18.0)	133 (18.0)	0.018 **	137 (15.0)	132 (18.0)	0.093	0.798	0.535
DBP mmHg)	77 (9.0)	73 (10.0)	0.002 *	76 (7.0)	73 (7.0)	0.046 *	79 (8.0)	75 (8.0)	0.004 *	78 (7.0)	78 (7.0)	0.950	0.399	0.099
HR (bpm)	71 (11.0)	69 (12.0)	0.279	69 (9.0)	67 (11.0)	0.115	74 (12.0)	70 (10.0)	0.010 **	73 (10.0)	71 (6.0)	0.467	0.417	0.268

Note: Intima–media thickness—left (IMT-L); systolic diameter—left (SD-L); diastolic diameter—left (DD-L); peak systolic velocity—left (PSV-L); end-diastolic velocity—left (EDV-L); Intima-media thickness—right (IMT-R); systolic diameter—right (SD-R); diastolic diameter—right (DD-R); peak systolic velocity—right (PSV-R); end-diastolic velocity—right (EDVR); systolic blood pressure (SBP); diastolic blood pressure (DBP); heart rate (HR) * Result obtained through T-Student test; ** result obtained through Wilcoxon test.

**Table 3 ijerph-19-03377-t003:** Biochemical marker results.

	AerG			IntG			ComG			CG			Time × Group (M1)	Time × Group (M2)
	M1 Mean (SD)	M2 Mean (SD)	Time (p)	M1 Mean (SD)	M2 Mean (SD)	Time (p)	M1 Mean (SD)	M2 Mean (SD)	Time (p)	M1 Mean (SD)	M2 Mean (SD)	Time (p)		
GLU (mg/dL)	94 (27)	89 (22)	0.006 **	95 (18)	90 (17)	0.041 **	98 (25)	95 (24)	0.309	97 (20)	99 (18)	0.449	0.728	0.112
TC (mg/dL)	184 (31)	185 (24)	0.935	174 (33)	172 (29)	0.650	182 (26)	178 (24)	0.346	176 (26)	177 (27)	0.757	0.512	0.381
HDL (mg/dL)	58 (14)	59 (12)	0.586	58 (15)	56 (15)	0.175	58 (11)	59 (13)	0.478	57 (11)	55 (11)	0.111	0.907	0.582
AI (mg/dL)	3.29 (0.63)	3.33 (0.49)	0.653	3.17 (0.54)	3.25 (0.6)	0.249	3.23 (0.68)	3.27 (0.71)	0.566	3.21 (0.64)	3.67 (0.52)	0.003 *	0.598	0.033 †
LDL (mg/dL)	106 (25)	108 (20)	0.712	95 (27)	95 (24)	0.855	103 (25)	100 (25)	0.498	97 (25)	97 (25)	0.989	0.338	0.226
TG (mg/dL)	102 (51)	101 (48)	0.903	105 (36)	104 (43)	0.682	107 (41)	104 (46)	0.503	116 (47)	119 (35)	0.478	0.393	0.191
MCP-1 (pg/mL)	141.74 (82.71)	82.45 (46.02)	0.001 **	80.43 (37.15)	63.42 (42.69)	0.059	231.79 (73.36)	193.44 (94.04)	0.033 **	72.76 (49.30)	54.34 (31.33)	0.093	0.000 †	0.000 †
MIP-1α (pg/mL)	93.09 (12.62)	91.04 (7.39)	0.009 **	96.19 (1.88)	96.45 (1.29)	0.179	110.49 (9.37)	110.43 (11.77)	0.214	90.90 (15.12)	90.96 (15.40)	0.546	0.000 †	0.000 †

Note: Glucose (GLU); cholesterol total (TC); high-density lipoprotein (HDL); atherogenic index (AI); low-density lipoprotein (LDL); triglycerides (TG); monocyte chemoattractant protein-1 (MPC-1); macrophage inflammatory protein (MIP-1α). * Result obtained through T-Student test; ** result obtained through Wilcoxon test; † results obtained through ANOVA.

## Data Availability

All data are shown in the manuscript.

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
