# Peer review of "The Impact of Aquatic Exercise Programs on the Intima-Media thickness of the Carotid Arteries, Hemodynamic Parameters, Lipid Profile and Chemokines of Community-Dwelling Older Persons: A Randomized Controlled Trial"

_ijerph, 2022, doi:10.3390/ijerph19063377_

Round 1

Reviewer 1 Report

The paper prepared by Farinha et al. presents a result of a study related to the impact of different programs of physical exercises on intima-media thickness and some other biomarkers of cardiovascular risk. This is a very important area for research because cardiovascular disease is one of the most important problems for public health worldwide. Such research may lead to improve in knowledge about cardiovascular disease prevention, what is very important. The study was generally well prepared. The research methodology as well as results are quite well described.

However, I believe that some significant changes to the manuscript are needed which may further improve the quality and relevance of the text. In brackets, I give the numbers of lines to which the comment relates.

It should be “vascular”, not “vas-Cular”. (3)

It should be “Randomized”, not “Random-Ized”. (4)

The following sentence should be written in another way, because it is not understandable for me: “As well as the chemokines for MCP-1 and MIP-1α in AerG and for MCP-1 in ComG.” (34-35) Perhaps, it should be a continue of the previous sentence, but it is written as a separate verbless sentence.

The introduction must be deeply changed in my opinion. The introduction is chaotic. It would be worth to divide the introduction into subsections, for example: the first related to general information about cardiovascular disease, the second related to markers used in cardiovascular assessment (mainly used in this study), the third related to the role of physical activity in the improvement of cardiovascular health, and the last in which the purpose of the present study would be precisely defined. Giving general information about cardiovascular disease it should be mentioned some information about atherosclerosis and about diseases which can develop in the course of atherosclerosis (coronary heart disease, stroke, but also peripheral arterial disease and its most severe form critical limb ischemia). You can use the recently published papers relating to most important information associated with some aspects of atherosclerotic cardiovascular diseases (DOI: 10.3390/ijerph17249339; DOI: 10.3390/ijerph182211970; DOI: 10.3390/cells10020226; DOI: 10.1038/s41586-021-03392-8).

“...” in the list of pro- and anti-inflammatory cytokines is inappropriate in my opinion. (91-92)

What does “non-institutionalized” mean in this context? (111)

“To analyzed the impact of different aquatic exercise programs on the intima-media thickness of the carotid arteries (IMT), blood pressure, lipid profile and chemokines (MCP-1 and MIP-1α) of older participants, three different physical exercise programs were used: continuous aerobic (AerG), interval aerobic exercise (IntG) and combined exercise (ComG).” – This sentence is in my opinion to improve. Maybe “IMT, blood pressure, lipid blood profile and chemokines blood concentration (MCP-1 and MIP-1α) were measured in all participants, undergoing three different physical exercise programs: continuous aerobic (AerG), interval aerobic exercise (IntG) and combined exercise (ComG).”? It is not appropriate to repeat in this place that the purpose of the study is to analyze the impact of exercise on cardiovascular disease biomarkers. In this place only methodology of the study should be precisely described. (113-117)

Number of references should be given in the text, not names of first authors. (135-136)

Did the Authors assess the level of physical activity of participants in the control group? I understand that they did only their normal activity in the study period, but was it assessed what is the level of physical activity in their normal lifestyle? Were they people generally active or with sedentary lifestyle?

More information about study population should be given, such as chronic diseases and medical history. There were only people without any chronic diseases and medication in the study population? If yes, it should be precisely written. If not, it is very important how many peoples in the study population were diagnosed with diabetes, hypertension, cardiovascular disease, how many people were after myocardial infarction, by-pass, etc.

In the figure 1. the letters are too small.

In my opinion, the figure 2. should be presented in “Results”, not in “Material and methods”. Moreover, it should be commented that although no significantly statistical differences in IMT were found in population, there were a participant with such high reduction in IMT. What is the interpretation of such phenomenon according to the Authors?

It should be rather “An example of ….. is presented in the Figure 2.”, not “Below is”. (193)

I suggest using the abbreviation “TG” for triglycerides and “TC” for total cholesterol, as usually used in the scientific and medical literature.

Although the discussion is generally well structured and it is at quite high scientific level, I think it would be worth do divide the discussion into subsections, for example the first related to IMT, the second related to blood pressure, the third related to lipid profile, and the last related to chemokines. It would make the discussion more attractive to read.

It should be “the biochemical”, not “thebiochemical”. (438)

Maybe “Similar results” would be more appropriate than “Identical results”. (449)

Strengths and limitations of the study must be discussed.

Reconsider the relevance of the title, please. In the title IMT is highlighted, so I was disappointed when I realized that no difference in IMT were found in the study. Maybe simply some biomarkers of cardiovascular disease should be mentioned in the title? Or influence of exercise on cardiovascular system? But of course, the final decision is for the Authors.

The list of references is prepared not in accordance with the rules required in papers published by MDPI.

English seems to be quite good (the text is generally understandable for me), but there are some sentences requiring improvement, which I mentioned above. I’m not English philologist and I feel not fully able to assess language quality.

Author Response

My colleagues and I would like to thank you for the opportunity to resubmit our manuscript to the International Journal of Environmental Research and Public Health. We found that the reviewer’ comments were very helpful and we have done our best to incorporate all of their suggestions and reply to the reviewer` comments. We believe that this has made a significant contribution to the overall quality of the manuscript.

We send have included an updated version of our manuscript with all the changes highlighted. If you require any additional information, please do not hesitate to get in touch with us.

Point 1: It should be “vascular”, not “vas-Cular”. (3)

Response 1: Thank you for your comment. Has been changed in the manuscript.

Point 2: It should be “Randomized”, not “Random-Ized”. (4)

Response 2:  Thank you for your comment. Has been changed in the manuscript.

Point 3: The following sentence should be written in another way, because it is not understandable for me: “As well as the chemokines for MCP-1 and MIP-1α in AerG and for MCP-1 in ComG.” (34-35) Perhaps, it should be a continue of the previous sentence, but it is written as a separate verbless sentence.

Response 3:  Thank you for your comment. Has been changed in the manuscript.

Point 4: The introduction must be deeply changed in my opinion. The introduction is chaotic. It would be worth to divide the introduction into subsections, for example: the first related to general information about cardiovascular disease, the second related to markers used in cardiovascular assessment (mainly used in this study), the third related to the role of physical activity in the improvement of cardiovascular health, and the last in which the purpose of the present study would be precisely defined. Giving general information about cardiovascular disease it should be mentioned some information about atherosclerosis and about diseases which can develop in the course of atherosclerosis (coronary heart disease, stroke, but also peripheral arterial disease and its most severe form critical limb ischemia). You can use the recently published papers relating to most important information associated with some aspects of atherosclerotic cardiovascular diseases (DOI: 10.3390/ijerph17249339; DOI: 10.3390/ijerph182211970; DOI: 10.3390/cells10020226; DOI: 10.1038/s41586-021-03392-8).

Response 4:  Thank you for your comment. The introduction has been rearranged in the manuscript.

Point 5: “...” in the list of pro- and anti-inflammatory cytokines is inappropriate in my opinion. (91-92)

Response 5:  Thank you for your comment. Has been changed in the manuscript.

Point 6: What does “non-institutionalized” mean in this context? (111)

Response 6:  Thank you for your comment. Means that the study sample consists of elderly people living in the community, that is, elderly people who do not live in institutions (for example: nursing homes).

Point 7: “To analyzed the impact of different aquatic exercise programs on the intima-media thickness of the carotid arteries (IMT), blood pressure, lipid profile and chemokines (MCP-1 and MIP-1α) of older participants, three different physical exercise programs were used: continuous aerobic (AerG), interval aerobic exercise (IntG) and combined exercise (ComG).” – This sentence is in my opinion to improve. Maybe “IMT, blood pressure, lipid blood profile and chemokines blood concentration (MCP-1 and MIP-1α) were measured in all participants, undergoing three different physical exercise programs: continuous aerobic (AerG), interval aerobic exercise (IntG) and combined exercise (ComG).”? It is not appropriate to repeat in this place that the purpose of the study is to analyze the impact of exercise on cardiovascular disease biomarkers. In this place only methodology of the study should be precisely described. (113-117)

Response 7:  Thank you for your comment. Has been changed in the manuscript.

Point 8: Number of references should be given in the text, not names of first authors. (135-136) 

Response 8:  Thank you for your comment. Has been changed in the manuscript.

Point 9: Did the Authors assess the level of physical activity of participants in the control group? I understand that they did only their normal activity in the study period, but was it assessed what is the level of physical activity in their normal lifestyle? Were they people generally active or with sedentary lifestyle?

Response 9:  Thank you for your comment. Although Physical activity levels were not assessed in the control group they were age matched persons that did not perform any type of regular systematic exercise. This information was added to the manuscript (164-165).

Point 10: More information about study population should be given, such as chronic diseases and medical history. There were only people without any chronic diseases and medication in the study population? If yes, it should be precisely written. If not, it is very important how many peoples in the study population were diagnosed with diabetes, hypertension, cardiovascular disease, how many people were after myocardial infarction, by-pass, etc.

Response 10:  Thank you for your comment. This information was added to the manuscript. (170-176)

Point 11: In the figure 1. the letters are too small. 

Response 11:  Thank you for your comment. Has been changed in the manuscript.

Point 12: In my opinion, the figure 2. should be presented in “Results”, not in “Material and methods”. Moreover, it should be commented that although no significantly statistical differences in IMT were found in population, there were a participant with such high reduction in IMT. What is the interpretation of such phenomenon according to the Authors?

Response 12: Thank you for your comment. The Figure 2 has been changed for the “Results” and the interpretation has been added to the manuscript (331-334).

Point 13: It should be rather “An example of ….. is presented in the Figure 2.”, not “Below is”. (193) 

Response 13: Thank you for your comment. Has been changed in the manuscript.

Point 14: I suggest using the abbreviation “TG” for triglycerides and “TC” for total cholesterol, as usually used in the scientific and medical literature. 

Response 14: Thank you for your comment. Has been changed in the manuscript. 

Point 15: Although the discussion is generally well structured and it is at quite high scientific level, I think it would be worth do divide the discussion into subsections, for example the first related to IMT, the second related to blood pressure, the third related to lipid profile, and the last related to chemokines. It would make the discussion more attractive to read. 

Response 15: Thank you for your comment. The discussion was structured according to your comment.

Point 16: It should be “the biochemical”, not “thebiochemical”. (438) 

Response 16: Thank you for your comment. Has been changed in the manuscript.

Point 17: Maybe “Similar results” would be more appropriate than “Identical results”. (449) 

Response 17: Thank you for your comment. Has been changed in the manuscript.

Point 18: Strengths and limitations of the study must be discussed. 

Response 18: Thank you for your comment. This information was added to the manuscript (525-537).

Point 19: Reconsider the relevance of the title, please. In the title IMT is highlighted, so I was disappointed when I realized that no difference in IMT were found in the study. Maybe simply some biomarkers of cardiovascular disease should be mentioned in the title? Or influence of exercise on cardiovascular system? But of course, the final decision is for the Authors. 

Response 19: Thank you for your comment. Has been changed in the manuscript.

Point 20: The list of references is prepared not in accordance with the rules required in papers published by MDPI. 

Response 20: Thank you for your comment. The list of reference has been changed in accordance with the rules required in papers published by MDPI.

Point 21: English seems to be quite good (the text is generally understandable for me), but there are some sentences requiring improvement, which I mentioned above. I’m not English philologist and I feel not fully able to assess language quality. 

Response 21: Thank you for your comment. We have taken into account your English improvement suggestions and checked the manuscript for other English mistakes.

Reviewer 2 Report

This manuscript focuses on the impact of aquatic exercise on community dwelling old persons. The authors tested the intima-media thickness of the carotid arteries and biomarkers of cardiovascular disease in the pre-intervention and post-intervention moments. Important implications are presented that aquatic physical exercise appears to improve cardiovascular health and have a beneficial effect in reducing important cardiovascular risk markers. This manuscript is well written and described. However, there are some minor errors needed to be revised:

Title:

“Cardio-vas-Cular”, “Random-Ized”. Fix these typos.

Abstract: 

Consider using the full name before the abbreviation or using the full name in abstract.

Introduction:

The authors should describe how the benefits and risks of aquatic exercise and how aquatic exercise may be the same and/or different as another physical exercises on the land.

Methods:

The resolutions of figure 1 is too low, “figure 2” isn’t mentioned in the main text, line193.

The systolic diameters (SD), diastolic diameters (DD), peak systolic velocity (PSV) and the endo-diastolic velocity (EDV) are independent parameters, the authors using the subtitle “2.3.2.Carotid Arteries Intima-Media Thickness” to describe is inappropriate, which is also found in abstract (describing them as “Regarding IMT”).

The duration is missed in IntG in figure 3.  

“Continuous aerobic (AerG): exercise aerobic (weeks 1-13, 60-65% maximum HR; weeks 14-28, 65-70% HR); Interval aerobic (IntG): exercise aerobic different intensities (weeks 1-13, 60-65% maximum HR interval to 70-75% maximum HR, weeks 14-28, 65-70% maximum HR interval to 75-80% maximum HR); Combined (ComG): exercise aerobic (weeks 1-13, 60%-65% maximum HR; weeks 14-28, 65-70% maximum HR)”, please describe why the authors set up the exercise programs at this specific intensity and duration.

Fix typos, line 221 etc.

Results:

Please add units in table 2 and 3.

Discussion:

it may be worth restating the characteristics and impact of aquatic exercise involving water.

Author Response

My colleagues and I would like to thank you for the opportunity to resubmit our manuscript to the International Journal of Environmental Research and Public Health. We found that the reviewer’ comments were very helpful and we have done our best to incorporate all of their suggestions and reply to the reviewer` comments. We believe that this has made a significant contribution to the overall quality of the manuscript.

We send have included an updated version of our manuscript with all the changes highlighted. If you require any additional information, please do not hesitate to get in touch with us.

Point 1: Title: “Cardio-vas-Cular”, “Random-Ized”. Fix these typos.

Response 1: Thank you for your comment. Has been changed in the manuscript.

Point 2: Abstract: Consider using the full name before the abbreviation or using the full name in abstract.

Response 2: Thank you for your comment. Has been changed in the manuscript.

Point 3: Introduction: The authors should describe how the benefits and risks of aquatic exercise and how aquatic exercise may be the same and/or different as another physical exercises on the land.

Response 3:  Thank you for your comment. This information was added to the manuscript. Line 126-129.

Point 4: Methods: The resolutions of figure 1 is too low, “figure 2” isn’t mentioned in the main text, line193.

Response 4:  Thank you for your comment. Has been changed in the manuscript.

Point 5: Methods: The systolic diameters (SD), diastolic diameters (DD), peak systolic velocity (PSV) and the endo-diastolic velocity (EDV) are independent parameters, the authors using the subtitle “2.3.2.Carotid Arteries Intima-Media Thickness” to describe is inappropriate, which is also found in abstract (describing them as “Regarding IMT”).

Response 5: Thank you for your comment. Has been changed in the manuscript.

Point 6: Methods: The duration is missed in IntG in figure 3. 

Response 6:  Thank you for your comment. This information was added to the Figure 3.

Point 7: Methods: “Continuous aerobic (AerG): exercise aerobic (weeks 1-13, 60-65% maximum HR; weeks 14-28, 65-70% HR); Interval aerobic (IntG): exercise aerobic different intensities (weeks 1-13, 60-65% maximum HR interval to 70-75% maximum HR, weeks 14-28, 65-70% maximum HR interval to 75-80% maximum HR); Combined (ComG): exercise aerobic (weeks 1-13, 60%-65% maximum HR; weeks 14-28, 65-70% maximum HR)”, please describe why the authors set up the exercise programs at this specific intensity and duration.

Response 7:  Thank you for your comment. The intensity of the exercise programs was defined according to the ACSM recommendations for the practice of physical exercise in the elderly population. Line 261-264. As for the duration of the programs, we wanted to evaluate the long-term effect of the different programs. 28 weeks was the duration of a complete sports season in the municipal swimming pool of Sertã. If we made it longer, we would have a huge reduction in the rate of adhesion to the different programs.

Point 8: Methods: Fix typos, line 221 etc.

Response 8:  Thank you for your comment. Has been changed in the manuscript.

Point 9: Results: Please add units in table 2 and 3.

Response 9:  Thank you for your comment. The units were added to tables 2 and 3.

Point 10: Discussion: It may be worth restating the characteristics and impact of aquatic exercise involving water.

Response 10:  Thank you for your comment. This information was added to the manuscript. Line 528-539.

Reviewer 3 Report

Dear authors, I have enjoyed reading your manuscript and I would like to congratulate you for the quality of the research design and the accurate reporting of procedures and data. 

I provide here below some comments with the final aim to improve some aspects of this manuscript. 

The introduction is quite clear and succinct but I felt more references concerning previous research on older adults could have been included to strengthen the need and rationale for this research. 

In the Methods section, the reference of the registration of the RCT in platforms such as clinical trials.org, if it has been performed, could be adequate. 

In participants and sample size I would like to know how participants were contacted and recruited (sampling method).

I have liked the clear presentation of the intervention program that helps a lot to understand the actions taken. Very useful. 

Results and discussion seem sufficient, clear and I consider these meet the standards of reporting RCT. I've only found that a more insightful paragraph with implications and translation of these results to clinical practice as well as potential/future research implications will just round off this section and the manuscript.
Finally, I am astonished to read that this highly complex and well-structured research has not received any specific funding to be carried out, so I reiterate once again my congratulations on the work accomplished and encourage researchers to continue with it.

Author Response

My colleagues and I would like to thank you for the opportunity to resubmit our manuscript to the International Journal of Environmental Research and Public Health. We found that the reviewer’ comments were very helpful and we have done our best to incorporate all of their suggestions and reply to the reviewer` comments. We believe that this has made a significant contribution to the overall quality of the manuscript.

We send have included an updated version of our manuscript with all the changes highlighted. If you require any additional information, please do not hesitate to get in touch with us.

Point 1: The introduction is quite clear and succinct but I felt more references concerning previous research on older adults could have been included to strengthen the need and rationale for this research.

Response 1: Thank you for your comment. The introduction has been improved during the review process.

Point 2: In the Methods section, the reference of the registration of the RCT in platforms such as clinical trials.org, if it has been performed, could be adequate.

Response 2: Thank you for your comment. The registration of the platform has not been carried out previously.

Point 3: In participants and sample size I would like to know how participants were contacted and recruited (sampling method).

Response 3:  Thank you for your comment. Participants were recruited through face-to-face invitation. We went to the installations of the municipal swimming pool of Sertã and made the invitation in person.

Point 4: I have liked the clear presentation of the intervention program that helps a lot to understand the actions taken. Very useful.

Response 4:  Thank you for your comment. This chapter was also improved during the review process, taking into account comments from other reviewers. 

Point 5: Results and discussion seem sufficient, clear and I consider these meet the standards of reporting RCT. I've only found that a more insightful paragraph with implications and translation of these results to clinical practice as well as potential/future research implications will just round off this section and the manuscript.

Response 5: Thank you for your comment. This information was added to the manuscript.

Point 6: Finally, I am astonished to read that this highly complex and well-structured research has not received any specific funding to be carried out, so I reiterate once again my congratulations on the work accomplished and encourage researchers to continue with it.

Response 6:  Thank you for your comment. Thank you very much for your words of encouragement. They are very important to us. Funding helps a lot for quality research, but when there are people with a good heart and goodwill, anything is possible.

Round 2

Reviewer 1 Report

The Authors responded satisfactorily to the suggestions presented in the review process. I believe that the manuscript has been significantly improved. While reading the revised version of the manuscript, I found a few minor flaws, mostly of an editorial nature. The work can be published after responding to the following minor recommendations:

Very well that the Authors added some information related to sex, medication, and cardiometabolic diseases, but in my opinion the information should be presented in table because it would be more elegant and easy to read. (239-246)

It is “endo-diastolic velocity”. I think it should be “end-diastolic velocity”. (281)

In the case of numerical values with a unit, there should be a space between the number and the unit. (I found such error in 105, 106, 107, 302, 416, 417)

Should be unified throughout the text, whether there should be spaces or not between the =, > or <, and the numbers.

It should be standardized whether the "." character is used as the decimal point or ",". According to me, in English should be ".".

Thank you very much for invitation to review this paper. Good luck for the Authors in further scientific work!

Reviewer 2 Report

Endorsed.